# UniErase: Towards Balanced and Precise Unlearning in Language Models

## Abstract

Large language models (LLMs) require iterative updates to address the outdated information problem, where LLM unlearning offers an approach for selective removal. However, mainstream unlearning methods primarily rely on fine-tuning techniques, which often lack precision in targeted unlearning and struggle to balance unlearning efficacy with general ability under massive and sequential settings. To bridge this gap, in this work, we introduce `UniErase`, a novel unlearning framework that demonstrates precision and balanced performances between knowledge unlearning and ability retaining. We first propose the Unlearning Token, which is optimized to steer LLMs toward a forgetting space. To achieve concrete unlearning behaviors, we further introduce the lightweight Unlearning Edit to efficiently associate the unlearning targets with this meta-token. Serving as a new unlearning paradigm via editing, `UniErase` achieves outstanding performances across batch, sequential, and precise unlearning tasks under fictitious and real-world knowledge scenarios. On the TOFU benchmark, compared with 8 baselines, `UniErase`, modifying only $\sim$ **3.66%** of the LLM parameters, outperforms the previous best-forgetting baseline by $\sim$ **4.01$\times$** for **model ability** with even higher unlearning efficacy. Similarly, `UniErase`, with better ability retention, also surpasses the previous best-retaining method by **35.96%** for **unlearning efficacy**, showing balanced and dual top-tier performances in the current unlearning community. We release our code at https://anonymous.4open.science/r/UniErase-5DE8.

## 1 Introduction

While the Large Language Models (LLMs) community (Guo et al., 2025; Chang et al., 2024; Wang et al., 2025a) has made significant advances in "learning" general abilities and domain-specific knowledge via pretraining and post-training (Kumar et al., 2025; Tie et al., 2025). Meanwhile, an equally crucial research direction is the complementary concept of LLM "unlearning" (Liu et al., 2025; Geng et al., 2025), which serves to address critical issues related to hallucination (Huang et al., 2025), privacy (Yan et al., 2024), and safety (Wang et al., 2025a)—including updating outdated knowledge, removing private information, and eliminating harmful contents (Lu et al., 2024; Zhang et al., 2024c; Xu, 2024). The core objectives of ideal unlearning is to enable LLMs, trained on trillion-token corpora, to only forget a specific data subset (the forgetting set) without compromising their general knowledge (the retaining set) and capabilities (Si et al., 2023; Maini et al., 2024).

Given the prohibitive computational cost of retraining LLMs from scratch while excluding the forgetting set, fine-tuning (FT) techniques has emerged as the predominant unlearning implementation (Maini et al., 2024; Yuan et al., 2024). Concretely, FT-based unlearning can be broadly categorized into two paradigms: **(I) Targeted unlearning** deliberately modifies LLMs' outputs of the forgetting set in *controlled* and *specified* manners, such as "I don't know"-like expressions (Wei et al., 2021; Rafailov et al., 2023); **(II) Untargeted unlearning** shifts the responses *away from* the original outputs but *without* specifying a particular direction, like irrelevant answers (Maini et al., 2024; Zhang et al., 2024b). These two paradigms both employ carefully designed loss functions with distinct objectives for the forgetting set (forgetting loss) and retaining set (retaining loss), respectively, thereby enabling knowledge erasure and retention (Yuan et al., 2024; Wang et al., 2025c;b).

However, fine-tuning inherently requires sufficient data volume to achieve effective optimization without overfitting, and the forgetting loss and retaining loss present competing objectives (Yuan

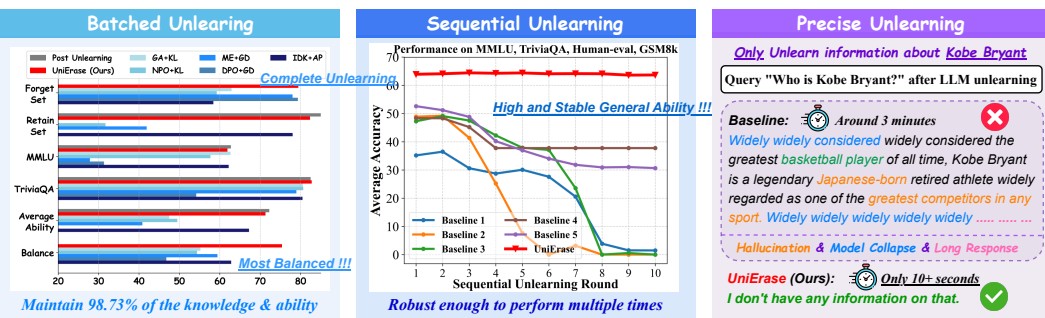

Figure 1: Our `UniErase` achieves the most balanced unlearning performances (*Left*) and maintains consistently high general capabilities (*Middle*), delivering rapid processing and high precision (*Right*).

et al., 2024; Geng et al., 2025). Besides, current FT-based unlearning, due to cost considerations, limits the retaining set to distributions near the forgetting set, which cannot represent the broad LLM knowledge (Maini et al., 2024). Consequently, these render two critical goals fundamentally challenging: ❶ **precise unlearning** for a small or even single-entry forgetting set, or ❷ **balanced unlearning** that concurrently preserves general abilities and knowledge while ensuring high unlearning efficacy on the forgetting set (Veldanda et al., 2024; Qu et al., 2024). Our empirical experiments across 8 baselines validate the second dilemma: for batch unlearning, the best-forgetting method loses **80.28%** of the general abilities, while the best-retaining baseline only forgets $\sim$ **half** of the target data.

In this paper, we aim to tackle these two critical issues for LLM knowledge unlearning, especially the balance challenge. To this end, we propose `UniErase`, a novel unlearning paradigm that balances unlearning efficacy and model abilities with dual-high performances, while supporting effective and efficient precise unlearning. Technically, `UniErase` consists of two innovative techniques: the **Unlearning Token** and the **Unlearning Edit (Udit)**. We first introduce the unlearning token that concretizes the concept of forgetting into a tangible entity and points to a representational space that encodes unlearning semantics. Specifically, the unlearning token directs the autoregressive prediction process to generate predefined forgetting responses for any input sequence that terminates with this token. To obtain it without affecting any other generation, we create and optimize a new meta token (Li & Liang, 2021; Lester et al., 2021) only in the embedding space of the LLM, with other parameters frozen. Building upon this, we further propose Udit, a data-volume-independent method (therefore supporting precise unlearning) that directly modifies model parameters to establish associations between the forgetting set and unlearning token, thus realizing unlearning via its directing property. More importantly, Udit employs the null space projection technique (Fang et al., 2024) to ensure parameter updates remaining orthogonal to the LLMs' existing knowledge representations, effectively preserving the retaining set and even general capabilities.

In contrast to FT-based unlearning, `UniErase` pioneers the modeling of LLM unlearning as a knowledge editing problem. We solve the problem that current LLM editing frameworks only support entity concept editing (Wang et al., 2024; Zhang et al., 2024a) via the unlearning token, and further propose Udit to truly achieve precise and balanced LLM unlearning. To validate the effectiveness of `UniErase`, following previous works (Yuan et al., 2024; Zhang et al., 2024b), we conduct extensive experiments on different scales of the `Llama-3` (Dubey et al., 2024) LLMs. Actually, with **8** baselines, we consider both fictitious and real-world knowledge in batch, sequential and precise unlearning scenarios (as illustrated in Figure 1). Evaluating via multi-dimensional metrics on the TOFU (Maini et al., 2024) benchmark, `UniErase` significantly outperforms the previous best-forgetting baseline, attaining $4.01\times$ performances in maintaining general knowledge and abilities while demonstrating better unlearning. Additionally, compared with the best-retaining baseline, `UniErase` preserves superior LLM abilities and is **35.96%** higher in unlearning efficacy.

In summary, our contributions can be listed as follows:

- **Brand-new Paradigm.** Our proposed `UniErase` represents a novel unlearning paradigm that exhibits outstanding performances by directly modifying LLM parameters instead of multi-round fine-tuning, significantly expanding the scopes of future research in the unlearning community.

- **Dual-high Balance.** `UniErase` achieves more thorough unlearning with better retention for general knowledge and abilities, boosting the practical usability of LLM unlearning.

- **Generalized Scenarios.** `UniErase` performs superbly across batch, sequential and especially precise unlearning for fictitious and real-world knowledge, covering diverse unlearning tasks.

## 2 RELATED WORKS

**Machine Unlearning.** The concept of machine unlearning (Bourtoule et al., 2021) from traditional models (Chen et al., 2022; Nguyen et al., 2022) is emerging as a rising research topic for LLMs (Liu et al., 2025; Thaker et al., 2024). Its primary goal is to enable LLMs to forget a subset $\mathcal{D}_f$ (e.g., privacy or harmful knowledge) of the training data $\mathcal{D}$ and maintain the knowledge on a retaining set $\mathcal{D}_r \subset \mathcal{D}$, without the high cost of retraining (Geng et al., 2025). Mainstream approaches relying on the fine-tuning techniques and designing various loss functions for different objectives to simultaneously forget $\mathcal{D}_f$ and retain $\mathcal{D}_r$. For example, GD (Liu et al., 2022) reduces the probability of generating outputs in $\mathcal{D}_f$ by ascending gradients, and introduces another loss to constrain the deviation. Meanwhile, NPO (Zhang et al., 2024b), inspired by preference optimization (Rafailov et al., 2023), realizes unlearning by solely using $\mathcal{D}_f$ as negative preferences, ignoring the positive terms. Other works, such as RMU (Huu-Tien et al., 2024) and LUNAR (Shen et al., 2025), employ steering-vector-like approaches (Cao et al., 2024) to forcibly modify hidden states and redirect $\mathcal{D}_f$ toward the inability space. Additionally, SPUL (Bhaila et al., 2024) makes preliminary attempts in unlearning by adding soft prompts (Li & Liang, 2021; Lester et al., 2021) during inference to manipulate model responses, but without modifying parameters to achieve essential forgetting.

**Model Edit.** LLMs may contain outdated, incorrect or even harmful information (Huang et al., 2025; Tonmoy et al., 2024). However, similar to unlearning, retraining for knowledge updates is costly, while fine-tuning overfits for precise scenarios. Thus, the model edit techniques (Wang et al., 2024; He et al., 2024) are proposed for truthfulness (Huang et al., 2024), and safety (Chen et al., 2024; Li et al., 2024). Early methods like ROME (Meng et al., 2022a) and MEMIT (Meng et al., 2022b) introduce the locate-then-edit paradigm by modifying the down-projection matrices in the LLMs' Multi-layer Perceptron (MLP) module. AlphaEdit (Fang et al., 2024) further preserves other knowledge via the null space projection operation. However, recent unlearning surveys like (Liu et al., 2025) have highlighted challenges including undefined edit objectives if directly applying editing for unlearning. In fact, editing itself targets at knowledge represented in the (subject, relation, object) triple formats and modifies the object to a new value (Meng et al., 2022a; Wang et al., 2024; Li et al., 2025), yet no single object token exists for the abstract concept of unlearning. Our `UniErase` addresses these fundamental issues by introducing Udit with the unlearning token.

## 3 PRELIMINARIES

**Notations.** We refer to an LLM with parameters $\theta$ as $\pi_\theta$. The target knowledge for the forgetting set and retaining set are represented as $\mathcal{D}_f$ and $\mathcal{D}_r$, respectively, where typical elements of both are question $q$ and answer $a$ pairs in the form of $d = (q, a)$. In addition, we denote the set of real numbers as $\mathbb{R}$, and the set of real number tensors with dimensions $(d_1, ..., d_n)$ as $\mathbb{R}^{d_1 \times ... \times d_n}$.

**Unlearning Target.** For an LLM $\pi_\theta$ trained with dataset $\mathcal{D}$, unlearning aims to make the model forget the contents in $\mathcal{D}_f$ as if it were trained solely on $\mathcal{D} \setminus \mathcal{D}_f$. In a parallel vein, unlearning must preserve the model's knowledge in $\mathcal{D}_r$ and even broader knowledge with general capabilities. Similar to the trade-off between harmless and helpfulness in LLM safety alignment (Varshney et al., 2023), unlearning involves a balance between the *unlearning efficacy* and *model ability*, formulated as:

$$\pi_\theta^* = \arg\max_{\pi_\theta} \mathbb{E}\Big[ \sum_{d \in \mathcal{D}_f} \text{Forget}(d; \pi_\theta) + \sum_{d \in \mathcal{D}_r} \text{Ability}(d; \pi_\theta)\Big], \tag{1}$$

where "Forget" and "Ability" are the standards or metrics for unlearning efficacy and model ability.

**Mainstream Unlearning Paradigms.** To achieve the goal in Eq. 1, current FT-based unlearning design diverse forgetting losses $l_f$ and retaining losses $l_r$, respectively, sometimes using the original model $\pi_\theta^{\text{ref}}$ as a reference. We unify their loss designs as follows, with $\beta$ and $\gamma$ as trade-off weights:

$$\arg\min_{\pi_\theta} = \beta \underbrace{\mathbb{E}_{(q,a) \sim \mathcal{D}_f}\big[l_f(q \mid a; \pi_\theta, \pi_\theta^{\text{ref}})\big]}_{\text{forgetting term}} + \gamma \underbrace{\mathbb{E}_{(q,a) \sim \mathcal{D}_r}\big[l_r(q \mid a; \pi_\theta, \pi_\theta^{\text{ref}})\big]}_{\text{retaining term}}. \tag{2}$$

In Eq. 2, the forgetting term is designed to make the model forget the contents on $\mathcal{D}_f$, while the retaining term aims to preserve the knowledge on $\mathcal{D}_r$. Current methods typically select $\mathcal{D}_r$ to be the neighboring knowledge of $\mathcal{D}_f$, which can not encompass diverse general knowledge and abilities. In Appendix B, we introduce the specific forms of various $l_f$ and $l_r$ in detail.

# 4 PERFORM UNLEARNING EDIT WITH UNLEANRING TOKEN

In this section, we first introduce the Unlearning Logical Chain to expound upon the fundamental principles of `UniErase` (▷ Section 4.1), as demonstrated in Figure 2. Then, we propose the unlearning token and elaborate on the techniques to obtain it via incorporating a minimal number of parameters (▷ Section 4.2). Subsequently, we introduce Udit to modify parameters for the unlearning targets (▷ Section 4.3), achieving balanced unlearning performances while supporting precise unlearning.

## 4.1 UNLEARNING LOGICAL CHAIN

Given an LLM $\pi_\theta$, for an input token sequence $q = [x_1 x_2 ... x_n]$, we assume the output token sequence is $a = [y_1 y_2 ... y_m]$. Then we abstract this generation process as a mathematical logic derivation:

$$x_1 x_2 ... x_n \xrightarrow{\pi_\theta} y_1 \xrightarrow{\pi_\theta} y_2 \xrightarrow{\pi_\theta} ... \xrightarrow{\pi_\theta} y_m, \tag{3}$$

where each $\xrightarrow{\pi_\theta}$ represents generating the next token based on all previously generated tokens.

***Proposition 1.*** The *Unlearning Token* (denoted as [UNL]) is a novel token, designed to direct the LLM's subsequent token generation to specific forgetting expressions. We refer to the token concatenation operator as $\oplus$. Then, for any $(q, a) \in \mathcal{D}$, [UNL] satisfies the following property:

$$x_1 x_2 ... x_n \oplus [\text{UNL}] \xrightarrow{\pi_\theta} y_{\text{idk}} \in \mathcal{D}_{\text{idk}} \quad \wedge \quad x_1 x_2 ... x_n \xrightarrow{\pi_\theta} a, \tag{4}$$

where operation $a \wedge b$ means that both $a$ and $b$ should be satisfied and $\mathcal{D}_{\text{idk}}$ contains different token sequences that express the semantics of forgetting or ignorance. Specifically, Eq. 4 stipulates that the newly acquired unlearning token should exclusively direct the model toward the forgetting semantic space when employed as a suffix, while preserving normal knowledge retrieval capabilities otherwise.

In Proposition 1, we have defined the [UNL] meta token. **However, when only $q$ is provided as input, the LLM still generates original normal responses rather than $y_{\text{idk}}$.** To realize unlearning, we need to modify model parameters so that: for any $q$, its next token prediction is [UNL], thereby internalizing *"forgetting $q$ with [UNL]"* as the LLM's inherent knowledge. To this end, we propose:

***Proposition 2.*** *Unlearning Editing* (Udit) modifies only a small set of parameters $\Delta\theta$, enabling the LLM to forget specified knowledge. For $\forall (q, a) \in \mathcal{D}_f$ and $\forall (q', a') \in \mathcal{D} \setminus \mathcal{D}_f$, Udit ensures that:

$$\left( |\Delta\theta| \ll |\theta|, \quad \theta \leftarrow \theta + \Delta\theta \quad \text{s.t.} \quad x_1 x_2 ... x_n \xrightarrow{\pi_\theta} [\text{UNL}] \right) \quad \wedge \quad \left( q' \xrightarrow{\pi_\theta} a' \right) \tag{5}$$

According to Eq. 5, Udit must demonstrate the ability to alter the subsequent token prediction of target unlearning contents to [UNL] via sparse parameter updates, while maintaining intact knowledge retrieval and response capabilities for non-target contents.

**Derivation:** Grounded in the aforementioned propositions, we establish the following *Unlearning Logical Chain*, which directly modifies LLM parameters to accomplish efficient targeted unlearning without compromising the model's retained knowledge and general capabilities:

$$\left( q' \xrightarrow{\pi_\theta} a' \right) \quad \wedge \quad \left( \theta \leftarrow \theta + \Delta\theta \quad \text{s.t.} \quad x_1 x_2 ... x_n \xrightarrow{\pi_\theta} [\text{UNL}] \xrightarrow{\pi_\theta} y_{\text{idk}} \in \mathcal{D}_{\text{idk}} \right) \tag{6}$$

This chain demonstrates the core spirits of `UniErase`, enabling us to realize unlearning on $\mathcal{D}_f$ via directing $a \in \mathcal{D}_F$ to $y_{\text{idk}}$, while preserving other untargeted generation $q' \to a'$.

## 4.2 UNLEARNING TOKEN

In this section, we present the specific techniques for deriving the unlearning token that fulfill the requirements in the Unlearning Logical Chain. In essence, the special token must satisfy three key criteria: redirecting arbitrary knowledge toward the forgetting space (Eq. 4), maintaining the normal response generation for other knowledge domains (Eq. 4), and being a generatable token (Eq. 5) rather than appearing only at the input end like prefix tuning (Li & Liang, 2021; Bhaila et al., 2024).

Figure 2: Paradigm of `UniErase` and comparison with mainstream FT-based unlearning.

### 4.2.1 OPTIMIZATION TO ATTAIN UNLEARNING TOKEN

Let $E \in \mathbb{R}^{n \times d} \subset \theta$ and $U \in \mathbb{R}^{n \times d}$ denote the embedding and unembedding matrices of the LLM $\pi_\theta$, respectively, where $n$ represents the vocabulary size and $d$ denotes the model dimension. We expand both $E$ and $U$ by incorporating two additional row vectors: $e_0 \in \mathbb{R}^d$ and $u_0 \in \mathbb{R}^d$ ($E \leftarrow E \cup e_0$ and $U \leftarrow U \cup u_0$), which correspond to the encoding and decoding representations for the unlearning token [UNL], respectively. Subsequently, we optimize the following objective to learn [UNL]:

$$\arg \min_{(e_u, \hat{e}_u)} \mathbb{E}_{(q,a) \sim \mathcal{D}_f, a' \sim \mathcal{D}_{\text{idk}}} \big[ -\log P(a' \oplus [\text{UNL}] \mid q \oplus [\text{UNL}]; \pi_\theta^u) - \log P(a \mid q; \pi_\theta^u) \big], \quad (7)$$

where $P(x \mid y; \pi_\theta)$ means the conditional probability of $\pi_\theta$ generating output $x$ when giving input $y$. Eq. 7 ensures that upon encountering [UNL], the model directs the original response $a$ (of input $q$) to the forgetting space, yielding $a' \in \mathcal{D}_{\text{idk}}$. Concurrently, the LLM is also forced to learn to generate $a' \oplus [\text{UNL}]$, satisfying the requirement specified in Eq. 5 that the LLM possesses the capability to generate [UNL]. Furthermore, since we only introduce additional parameters in $E$ and $U$, while keeping all other parameters frozen, the aforementioned [UNL] optimization does not interfere with normal response generation for other knowledge domains.

### 4.2.2 ROBUSTNESS ENHANCEMENT OF UNLEARNING TOKEN

**Parameter Robustness.** In the Unlearning Logical Chain, $\Delta$ in Udit may render the previously learned unlearning token ineffective, so we need to enhance its robustness against slight parameter perturbations. While directly incorporating constraints into Eq. 7 to achieve this is challenging, we leverage the fact that Udit's target parameters are confined to the down projection matrices $W_{dp}$ within the MLP module. Therefore, in the optimization process of Eq. 7, we introduce the following perturbations in the LLM to improve the parameter robustness of the resulting unlearning tokens:

$$W_{\text{dp}}^i \in \theta \leftarrow W_{\text{dp}}^i + \alpha f(W_{\text{dp}}^i) \cdot W_{\text{dp}}^i, \quad (8)$$

where parameter $\alpha$ controls the intensity and $f$ is a function mapping $W_{\text{dp}}^i$ to a scalar.

### 4.3 UNLEARNING EDIT

With [UNL] obtained, we propose Udit to bridge it with the knowledge to be unlearned, ensuring the internalization of unlearning targets. Following previous model editing techniques (Meng et al., 2022a;b), we target at the down projection matrices $W_{\text{dp}}^i$ in the MLP module. As shown in Figure 2, the $i$-th MLP layer performs the following computation ($\sigma$ and $\gamma$ are activation functions):

$$h^i = h^{i-1} + a^i + m^i, \quad \underbrace{m^i}_{v} = W_{\text{dp}}^i \underbrace{\sigma(W_{\text{up}}^i \gamma(h^{i-1} + a^i))}_{k}, \quad (9)$$

where $h^i$, $a^i$, and $m^i$ represent the hidden state, attention and MLP output in the $i$-th layer, respectively. Udit exclusively updates $W_{\text{dp}}^i \leftarrow \tilde{W}_{\text{dp}}^i$ to satisfy the new association $\tilde{W}_{\text{dp}}^i k^* = v^*$, where $k^*$

corresponds to the hidden state of targeted knowledge $q \in \mathcal{D}_f$, and $v^*$ is optimized to maximize the prediction probability of [UNL] as the next token when the input is $q$. In other words, Udit builds new knowledge mappings from the original $W_{dp}^i k = v \rightarrow W_{dp}^i k^* = v^*$ to ensure the next token prediction for $q \in \mathcal{D}_f$ is modified to [UNL] (Eq. 5). The detailed procedures for obtaining these $k^*$ and $v^*$ for each knowledge-answer pair $(q, a)$ are provided in Appendix C.

We construct the unlearning matrices by stacking the corresponding key and value vectors. Specifically, for each $(q, a) \in D_f$ to be unlearned, we stack their corresponding unlearning key vectors $k^*$ and value vectors $v^*$ into matrices $(K_f, V_f)$, respectively. Similarly, for each $(q, a) \in D_r$ to be retained, compile their normal key and value vectors into matrices $(K_r, V_r)$.

Then, we propose the core technique of Udit: by updating $W_{dp}^i \leftarrow W_{dp}^i + \Delta^*$, we construct new mappings between $q \in \mathcal{D}_f$ and [UNL], while preserving the retrieval of other knowledge ($q \in \mathcal{D}_r$) to approximate the unlearning objective described in Eq. 1. The parameter update $\Delta^*$ is optimized via:

$$\Delta^* = \arg\min_{\Delta} \big( \underbrace{\|(W_{dp} + \Delta)K_f - V_f\|^2}_{\text{forget term}} + \underbrace{\|(W_{dp}^i + \Delta)K_r - V_r\|^2}_{\text{retain term}} \big). \tag{10}$$

In Eq. 10, for all $(q, a) \in \mathcal{D}_f$, the forget term modifies the first token of the response $a$ to [UNL], while the retain term ensures that all $(q, a) \in \mathcal{D}_r$ retain their original input-output pairs. Through mathematical derivation (provided in Appendix C), we can *quickly* get its **closed-form solution**:

$$\Delta^* = (V_f - W_{dp}K_f)K_f^T(K_rK_r^T + K_fK_f^T)^{-1}. \tag{11}$$

Notably, Eq. 11 accommodates a significantly broader $\mathcal{D}_r$ than FT-based unlearning methods, thereby preserving a wider range of general knowledge and capabilities. To this end, we include a general knowledge dataset $\mathcal{D}_g$ within $\mathcal{D}_r$ (with $|\mathcal{D}_g| \gg |\mathcal{D}_f|$), which remains computationally infeasible for other unlearning approaches (Yuan et al., 2024; Zhang et al., 2024b; Maini et al., 2024).

**Null-space Projection Unlearning.** Inspired by AlphaEdit (Fang et al., 2024), we further optimize Udit into a null space projection formulation to further reduce the impact of unlearning on general knowledge. Specifically, we obtain the new parameter update $\Delta P$ by right-multiplying with matrix $P$, which projects $\Delta$ onto the null space of $K_r$ such that $\Delta P K_r = 0$. Through straightforward calculation, the retain term in Eq. 10 degenerates to 0 (meaning no influence on the retaining set), thus we only need to optimize the forget term. The new optimization objective can be formulated as:

$$\Delta^* = \arg\min_{\Delta} \big( \underbrace{\|(W_{dp} + \Delta P)K_f - V_f\|^2}_{\text{forget term}} + \underbrace{\|\Delta P\|^2}_{\text{constrain term}} \big), \tag{12}$$

where the constraint term is incorporated to limit the magnitude of parameter updates. Through a similar derivation as presented in Eq. 11, we arrive at the following closed-form solution:

$$\Delta^* = (V_f - W_{dp}K_f)K_f^T P(K_fK_f^T P + I)^{-1}. \tag{13}$$

The complete mathematical derivation of the closed-form solution presented in Eq. 13, as well as the methodology for computing $P$, is provided in Appendix C.

## 5 EXPERIMENT

In this section, we experimentally validate and analyze the effectiveness of our balanced and precise `UniErase` in the following three scenarios: **(I) Batch Unlearning** ($\triangleright$ Section 5.2) refers to making an LLM forget a large forgetting dataset in a single unlearning step. **(II) Sequential Unlearning** ($\triangleright$ Section 5.3) performs multiple rounds of unlearning tasks, testing whether unlearning methods cause LLMs to collapse for consecutive scenarios. **(III) Precise Unlearning** ($\triangleright$ Section 5.4) considers extremely small (single-entry) forgetting sets to test the precision of unlearning methods.

### 5.1 OVERALL SETTINGS

**Datasets & Models.** We consider two widely adopted TOFU (Maini et al., 2024) and RETURN (Liu et al., 2024) benchmarks for fictitious and real-world knowledge unlearning, respectively. They both contain serval forgetting sets and corresponding and neighboring retaining sets. Highlighting

Table 1: **Batch unlearning performances of different unlearning methods for the TOFU-inject Llama-3.1-8B-Instruct.** "Base" means the original LLM before unlearning. In each row, we **bold** the maximum value and underline the second largest one. "Forget" and "Retain" refer to the $\mathcal{D}_f$ and $\mathcal{D}_r$ datasets in TOFU, while "Real" is the real fact test dataset in TOFU.

| Model / Category | | | Untargeted Unlearning (UU) | | | | | Targeted Unlearning (TU) | | | |
|---|---|---|---|---|---|---|---|---|---|---|---|
| tofu_Llama-3.1-8B-Instruct_full | | | GA+GD | GA+KL | NPO+GD | NPO+KL | ME+GD | DPO+GD | DPO+KL | IDK+AP | UniErase |
| Dataset | Metric | Base | NIPS24 | - | COLM24 | | ICLR25 | COLM24 | - | ICLR25 | (Ours) |
| **Forget** | FE | 10.95 | 58.29 | 62.91 | 58.31 | 59.24 | 78.01 | 79.31 | 79.02 | 58.42 | **79.43** |
| **Retain** | RE | 86.34 | 27.47 | 0.00 | 43.38 | 31.73 | 41.92 | 0.00 | 0.00 | 78.03 | **82.32** |
| **Real** | RE | 76.44 | 42.75 | 0.00 | 53.88 | 46.75 | 57.63 | 0.00 | 0.00 | 74.73 | **75.18** |
| **MMLU** | Acc | 62.75 | 62.18 | **62.66** | 44.30 | 57.69 | 27.85 | 31.34 | 19.73 | 62.18 | 61.89 |
| | Idk | 0.00 | 0.00 | 0.00 | 0.00 | 0.00 | 0.00 | 51.07 | **69.80** | 0.00 | 0.00 |
| | Len | 8.55 | 20.14 | 172.8 | **511.8** | 499.7 | 28.41 | 7.03 | 7.41 | 6.32 | 8.68 |
| **TriviaQA** | Acc | 82.49 | 82.22 | 80.53 | 82.44 | 80.66 | 78.97 | 54.17 | 35.81 | 80.47 | **82.75** |
| | Idk | 0.00 | 0.00 | 0.00 | 0.00 | 0.00 | 0.00 | 26.89 | **50.46** | 0.00 | 0.00 |
| | Len | 9.53 | 13.77 | 43.24 | **512.0** | 492.0 | 27.44 | 7.87 | 7.85 | 7.96 | 9.53 |
| **Human-Eval** | Acc | 56.10 | 54.27 | **64.02** | 0.07 | 23.78 | 0.00 | 0.00 | 0.00 | 48.78 | 54.27 |
| | Idk | 0.00 | 0.00 | 0.00 | 0.00 | 0.00 | 0.00 | 72.57 | **85.98** | 0.00 | 0.00 |
| | Len | 61.53 | 66.85 | 88.46 | **316.6** | 205.7 | 18.91 | 22.26 | 15.36 | 60.74 | 61.98 |
| **GSM8k** | Acc | 69.37 | 75.36 | **77.71** | 53.53 | 56.33 | 38.59 | 0.00 | 0.00 | 59.14 | 71.57 |
| | Idk | 0.00 | 0.00 | 0.00 | 0.00 | 0.00 | 0.00 | **100.0** | **100.0** | 0.00 | 0.00 |
| | Len | 99.48 | 147.7 | 189.7 | **511.6** | 468.3 | 97.15 | 8.00 | 8.00 | 72.38 | 100.4 |
| **Retain Average (RA)** | | 72.25 | 57.38 | 47.49 | 46.27 | 49.49 | 40.83 | 14.25 | 9.26 | 67.22 | **71.33** |
| **Retain Ratio (%)** | | 100.0 | 79.41 | 65.73 | 64.04 | 68.50 | 56.51 | 19.72 | 12.81 | 93.04 | **98.73** |
| **Balance = (FE+RA)/2** | | 41.60 | 57.83 | 55.20 | 52.29 | 54.37 | 59.42 | 46.78 | 44.14 | 62.82 | **75.38** |

the retention for general abilities, we employ MMLU (Hendrycks et al., 2020) for fact answering, TriviaQA (Joshi et al., 2017) for context comprehension, GSM8k (Cobbe et al., 2021) for math reasoning, and Human-Eval (Chen et al., 2021) for coding. Following previous works (Maini et al., 2024; Yuan et al., 2024), we perform unlearning on the Llama-3.1-8B-Instruct and Llama-3.2-3B-Instruct (Touvron et al., 2023). For fictitious knowledge unlearning, we apply the model versions[1] fine-tuned on TOFU. More details of these datasets and models are demonstrated in Appendix D.

**Metrics.** We consider multiple metrics to comprehensively evaluate performances on unlearning and retention. For unlearning efficacy, in line with prior research (Maini et al., 2024; Zhang et al., 2024b; Yuan et al., 2024), we employ ROUGE (word-level match), Probability (ground truth likelihood), Truth Ratio (correct-incorrect preference), Token Entropy (generation diversity), Similarity (semantic similarity), and Entailment Score (factual correctness) (Ferrández et al., 2008). To obtain an integrated indicator, we calculate the arithmetic mean of these metrics on $\mathcal{D}_f$ as the overall *Forgetting Efficacy (FE)* (Yuan et al., 2024). For neighboring knowledge retention on $\mathcal{D}_r$, we similarly apply the above metrics and compute their harmonic mean to be the *Retaining Efficacy (RE)* (Yuan et al., 2024). Besides, for general abilities, we report the accuracy (Acc), "I do not know"-like response ratio (Idk), and average response token number (Len). Besides, *Retaining Average (RA)* is the mean of RE and all Accs, as the final metric for retention. We provide the details of these metrics in Appendix E.

**Baselines.** To demonstrate the effectiveness of our proposed paradigm, we evaluate `UniErase` against 8 FT-based unlearning baselines. Our comparison includes four primary forgetting losses: GA (Liu et al., 2022; Yao et al., 2024), DPO (Maini et al., 2024), NPO (Zhang et al., 2024b), and IDK (Yuan et al., 2024). These forgetting losses are combined with various retaining loss functions, including KL (Maini et al., 2024), GD (Liu et al., 2022), and ME (Yuan et al., 2024), resulting in the following baseline configurations: GA+GD, GA+KL, NPO+GD, NPO+KL, ME+GD, DPO+GD, DPO+KL, and IDK+AP. The specific parameter settings of all methods are detailed in Appendix F.

## 5.2 BATCH UNLEARNING

We perform batch unlearning on the TOFU and RETURN forgetting datasets, eliminating 400 fictitious and real-world knowledge entries in a single batch operation. The experimental results are presented in Table 1 and Figure 3. We provide supporting results of another LLM in Appendix G.

**Obs. 1: `UniErase` achieves dual-high, near-lossless and the most balanced unlearning performances, preserving 98.73% of LLMs' general abilities.** As shown in Table 1, `UniErase` attains the highest FE of 79.43 on $\mathcal{D}_f$, outperforming all FT-based baselines. Concurrently, it attains an RE of 82.32 on $\mathcal{D}_r$, surpassing the second-best method (IDK+AP) by 4.29, while `UniErase`'s FE

---

[1]https://huggingface.co/open-unlearning/tofu_Llama-3.1-8B-Instruct_full

is significantly higher by 35.96%. Regarding general capabilities, `UniErase` demonstrates superior performance, achieving the highest and second-highest accuracy in comprehension and coding tasks, respectively. For MMLU reasoning, it incurs only a 1.37% performance drop, matching with the best baselines (GA+KL, IDK+AP). From a holistic evaluation perspective encompassing both forgetting and retaining, `UniErase` wins the highest balance score of 78.38, which is $1.15\times$ and $1.71\times$ higher than the second-best and worst-performing methods, respectively. Notably, according to Figure 3, these observations also hold true on RETURN benchmark, validating the effectiveness of `UniErase`.

**Obs. 2: `UniErase` is *entirely* immune to the over-unlearning problem.** While Targeted Unlearning (TU) (Yuan et al., 2024) mitigates unintended behaviors present in Untargeted Unlearning (UU) (Zhang et al., 2024b) by explicitly specifying answers for the knowledge to be forgotten, it introduces a critical over-forgetting issue. As demonstrated in Table 1, all UU baselines maintain normal response patterns across the four general ability datasets, consistently achieving Idk = 0. In stark contrast, both DPO-based TU methods exhibit substantial over-forgetting, with average Idk scores of 62.63 and 76.56, respectively. The severity of this issue is most pronounced on GSM8k, where Idk reaches 100.0. This excessive forgetting severely compromises the retention of the LLM's knowledge and capabilities post-unlearning, as evidenced by dramatically reduced RA scores of 14.25 and 9.26. Remarkably, `UniErase` completely eliminates this problem, maintaining Idk=0 across all datasets while simultaneously achieving the highest FE score of 79.43 among all baselines.

**Obs. 3: `UniErase` does not trigger unexpected behaviors such as inflated response length.** The preceding discussion underscores the issue of unintended behaviors in UU methods, and Table 1 provides concrete evidence of this phenomenon through response length analysis. For the four datasets evaluating general capabilities, we impose a maximum generation length of 512 tokens. While TU methods (including our `UniErase`) maintain response lengths comparable to the base model—with average token counts on MMLU ranging between 6.32 and 8.68—all UU methods demonstrate varying degrees of response length inflation. The most pronounced cases involve the two NPO-based methods, where NPO+GD generates responses up to $50\times$ longer than the base model on MMLU according to the Len metric, while paradoxically experiencing performance degradation ($62.75\rightarrow44.3$). This indicates that UU baselines consistently generate responses that reach the maximum token limit by padding with uninformative contents.

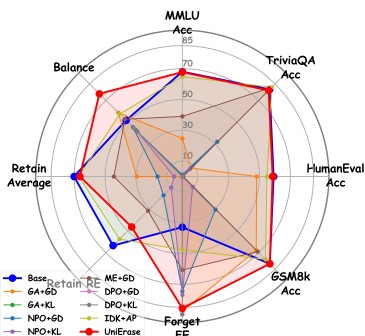

Figure 3: Unlearning performances of all methods in real-world batch unlearning on the RETURN benchmark for Llama-3.1-8B-Instruct.

## 5.3 Sequential Unlearning

Sequential unlearning scenarios evaluate the robustness of unlearning methods by testing whether they cause forgetting performance degradation and model collapse (Yuan et al., 2024). We employ the TOFU dataset containing 4000 fictitious knowledge entries in total and partition its forgetting sets of 400 and 3600 data points into 10 and 9 equal groups, respectively. Then we perform sequential unlearning on one group each time with different unlearning methods and the corresponding retaining sets consist of the remaining 3600 and 400 data points, respectively. The results for Llama-3.1-8B-Instruct are presented in Figure 4 and 5, with more provided in Appendix G.

**Obs. 4: `UniErase` exhibits exceptional stability for continuous LLM unlearning while preserving model capabilities.** As illustrated in the middle of Figure 4, the blue baselines achieve higher FE across multiple rounds; however, the left section reveals this comes at a substantial cost to general capabilities—with performances dropping to approximately 25.0 (DPO+KL, DPO+GD) or even 0 (GA+GD, GA+KL). Conversely, the green baselines and our `UniErase` exhibit moderately lower per-round FE scores but preserve significantly more knowledge and capabilities, maintaining Balance scores of approximately 55.0 and 75.0, respectively. Notably, `UniErase` consistently outperforms the green baselines across all metrics while sustaining this balance. On average (light dashed line), `UniErase` achieves a RA score that is $1.5\times$ to $1.8\times$ higher, while its FE exceeds the green baselines by 14.29%, showing dual-high and more balanced unlearning performances. Furthermore, Figure 5 demonstrates that `UniErase` achieves the highest MMLU accuracy with minimal variance, reinforcing this observation when scaling the sequential batch from $40 \rightarrow 400$.

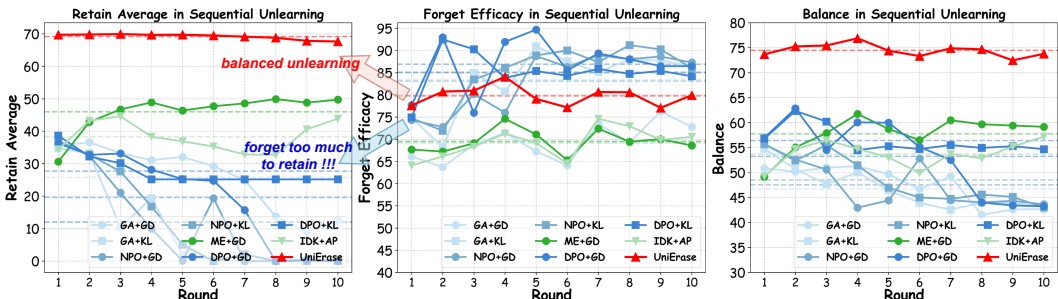

Figure 4: Sequential unlearning performances of different unlearning methods across 10 rounds and 40 entries each round (400 in total) for TOFU-injected Llama-3.1-8B-Intruct.

Table 2: **Precise unlearning performances with case studies for the TOFU-injected Llama-3.1-8B-Instruct.** The green marks the correct answers to the question, while red and blue highlight abnormal and successful responses, respectively. Maximum values in each column are in **bold**. Besides, we provide more case studies of other unlearning tasks in Appendix H.

| Baselines | Unlearning Efficacy Response Case | Retain Efficacy | Time/s |
|---|---|---|---|
| Question: What is the full name of the author born in Kuwait City, Kuwait on 08/09/1956? | | | |
| Answer: The full name of ... is Basil Mahfouz Al-Kuwaiti.  – Model: tofu_Llama-3.1-8B-Instruct_full | | | |
| GA+GD | The author author ... is named Leila Al-Sabah. | 71.55 | ∼165 |
| GA+KL | The author author born on ... is named Leila Al-Sabah. | 71.49 | ∼173 |
| NPO+GD | The author born in ... is named Akbar S. Ahmed. | 69.71 | ∼174 |
| NPO+KL | The author born in ... is named Akbar Al-Sabah. | 69.67 | ∼177 |
| ME+GD | f o o | 73.28 | ∼168 |
| DPO+GD | The ... in Kuwait City, Kuwait on 08/09/1956 is not provided. | 72.92 | ∼189 |
| DPO+KL | The ... in Kuwait City, Kuwait on 08/09/1956 is not provided. | 72.94 | ∼192 |
| IDK+AP | I've got no idea about that. | 72.84 | ∼180 |
| UniErase | That's beyond my current knowledge base. | **73.63** | **∼12** |

Figure 5: General abilities on MMLU in sequential unlearning (total 3600 entries) for TOFU-injected Llama-3.1-8B-Intruct.

## 5.4 PRECISE UNLEARNING

Given the huge computational overhead of FT-based unlearning when executing precise unlearning at scale—where each target knowledge requires fine-tuning on the full $\mathcal{D}_r$ containing 3600 data points—we randomly sampled 20 knowledge items from TOFU as the individual unlearning targets. In Table 2, with representative case studies, we report the average Retain Efficacy and time cost.

**Obs. 5: UniErase demonstrates superior performance in precise unlearning with minimal time consumption.** As shown in Table 2, among the UU baselines, the post-unlearning LLMs exhibit hallucination and model collapse phenomena. Specifically, the GA-based and NPO-based baselines generate incorrect names (Leila Al-Sabah) in their responses, while ME+GD leads to complete model collapse, producing nonsensical character outputs. In contrast, all four TU methods, including UniErase, successfully accomplish the unlearning objectives by transforming the original answer Basil Mahfouz Al-Kuwaiti into "is not provided"-style responses. UniErase further distinguishes itself by the highest RE score of 73.63 and requiring substantially lower computational overhead—completing the unlearning task in less than $\frac{1}{10}$ the time required by other baselines.

## 6 CONCLUSION

In this work, we propose UniErase, a novel paradigm for LLM unlearning that operates by directly modifying internal model parameters. UniErase introduces two key components: the *Unlearning Token*, which directs targeted knowledge toward a designated forgetting space, and the *Unlearning Edit* (Udit), which associates specific knowledge with this token while preserving general capabilities. Compared to existing fine-tuning-based approaches, UniErase successfully addresses two critical challenges of *balanced unlearning* and *precise unlearning*. To evaluate our paradigm, employing Llama family LLMs, we compare against 8 baseline methods and provide a comprehensive assessment of post-unlearning model performance on 4 general capability datasets. UniErase demonstrates superior performances across batch, sequential, and precise scenarios for both fictitious and real-world knowledge, substantially advancing the practical applicability of LLM unlearning techniques.

ETHICS STATEMENT

This work presents fundamental machine learning research. We have carefully considered its ethical implications and confirm that this study adheres to the ICLR Code of Ethics. The data used consists of publicly available or ethically compliant benchmark datasets. Potential societal impacts of the research are discussed in Section 1.

REPRODUCIBILITY STATEMENT

To facilitate the reproducibility of this research, we have provided necessary details in the appendices. This includes sufficient descriptions of the experimental setup (Appendix F), key implementation details of our methods (Appendix C), and essential information of used datasets (Appendix D). Relevant code and resources supporting the findings of this paper is publicly available in the anonymous code base mentioned in the abstract.

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

## A    FUTURE WORKS

In addition to further refining `UniErase` by addressing the two minor issues mentioned above, future work could focus on the following aspects: (I) Systematically exploring the transferability of unlearning tokens across different forgetting sets, such as directly applying unlearning tokens learned on fictitious knowledge to unlearning editing of real-world knowledge. Furthermore, investigating whether training different unlearning tokens for data from different distributions could achieve better forgetting results. (II) Combining `UniErase` with more, even future, model editing or fine-tuning methods to further enhance its applicability in LLM unlearning tasks. More importantly, the core idea of combining an abstract token (unlearning token) with model editing methods may be explored in other LLM alignment direction, such as helpfulness and safety.

## B    UNLEARNING LOSSES

In this section, we provide a detailed introduction to the losses used in previous fine-tuning-based unlearning methods (which also serve as baselines in the experiments) with their forgetting losses $\mathcal{L}_f$ and the knowledge retaining losses $\mathcal{L}_r$. We denote the forgetting set as $\mathcal{D}_f$, the retaining set as $\mathcal{D}_r$, and "I do not know"-like ignorant expressions as set $\mathcal{D}_{\text{idk}}$.

**Forgetting Loss 1: Gradient Ascent (GA):**

$$\mathcal{L}_{\text{GA}}(\mathcal{D}_f; \pi_\theta) = -\mathbb{E}_{(q,a)\sim\mathcal{D}_f}\left[-\log p(q \mid a; \pi_\theta)\right]. \tag{14}$$

Eq 14 is one of the simplest and straighforward methods for untargeted unlearning. Instead of minimizing the loss like in training or fine-tuning, GA does the opposite—it maximizes the loss on $\mathcal{D}_f$. Mathematically, it updates the model parameters $\theta$ to increase the prediction loss $l(y|x;\theta)$ for $\mathcal{D}_f$, effectively "unlearning" the associated patterns.

**Forgetting Loss 2: "I Do not Know" Optimization (IDK):**

$$\mathcal{L}_{\text{IDK}}(\mathcal{D}_f, \mathcal{D}_{\text{idk}}; \pi_\theta) = \mathbb{E}_{q\sim\mathcal{D}_f, a\sim\mathcal{D}_{\text{idk}}}[-\log p(a \mid q; \pi_\theta)] \tag{15}$$

Eq 15 redefines machine unlearning by framing it as an instruction-tuning task. Instead of directly removing unwanted data like GA, it relabels queries in $\mathcal{D}_f$ with randomized rejection responses (e.g., "I don't know") drawn from a predefined collection $\mathcal{D}_{\text{idk}}$ containing 100 such templates.

**Forgetting Loss 3: Direct Preference Optimization (DPO):**

$$\mathcal{L}_{\text{DPO}}(\mathcal{D}_f; \pi_\theta) = \mathbb{E}_{(q,a_w)\sim\mathcal{D}_f, a_l\sim\mathcal{D}_{\text{idk}}}\left[\log \sigma\left(\beta \log \frac{\pi_\theta(a_w \mid q)}{\pi_\theta^{\text{ref}}(a_w \mid q)} - \beta \log \frac{\pi_\theta(a_l \mid q)}{\pi_\theta^{\text{ref}}(a_l \mid q)}\right)\right], \tag{16}$$

where $a_w$ and $a_l$ are the original and "I do not know"-like responses, respectively. Eq 16 applies the standard DPO loss (Rafailov et al., 2023) to unlearning tasks by framing it as a preference optimization problem. Specifically, it treats answers from $\mathcal{D}_f$ as negative (undesired) samples and pairs them with rejection templates from $\mathcal{D}_{\text{idk}}$ as positive (preferred) samples. This contrastive approach fine-tunes the model to align responses away from $\mathcal{D}_f$ while reinforcing desired behaviors through ignorance-based guidance.

**Forgetting Loss 4: Negative Preference Optimization (NPO):**

$$\mathcal{L}_{\text{NPO}}(\mathcal{D}_f; \pi_\theta) = -\frac{2}{\beta}\mathbb{E}_{(q,a)\sim\mathcal{D}_f}\left[\log \sigma\left(-\beta \log \frac{p(a \mid q; \pi_\theta)}{p(a \mid q; \pi_\theta^{\text{ref}})}\right)\right]. \tag{17}$$

Eq 17 is an adaptation of Eq 16 that also frames unlearning as a preference optimization task. Unlike DPO, which balances both preferred and dispreferred responses, NPO specifically targets undesired outputs by treating samples from $\mathcal{D}_f$ as negative (non-preferred) examples. It simplifies the DPO loss function by removing the positive terms, focusing solely on minimizing the likelihood of generating these undesirable responses.

**Forgetting Loss 5: Maximizing Entropy (ME):**

$$\mathcal{L}_{\text{ME}}(\mathcal{D}_f; \theta) = \mathbb{E}_{(q,a)\sim\mathcal{D}_f}\left[\frac{1}{T}\sum_{t=1}^{T} \text{KL}(P_t \parallel U_{[K]})\right], \tag{18}$$

where $P_t = p(a'_t | a'_{<t}; \pi_\theta)$ is the predicted probability for the $t$-th token in $a' = a \circ q$ and $\mathcal{U}_{[K]}$ is a uniform distribution over the vocabulary of size $K$, where each value is $1/K$. Eq 18 aligns the LLM's predictions on $\mathcal{D}_f$ with those of a randomly initialized model, which inherently lacks knowledge of the data. Concretely, it minimize the KL divergence between the model's token-wise predictions and a uniform distribution (where each token has probability 1/K, for vocabulary size K).

**Retaining Loss 1: Gradient Descent (GD):**

$$\mathcal{L}_{\text{GD}}(\mathcal{D}_r; \pi_\theta) = \mathbb{E}_{(q,a)\sim\mathcal{D}_r} \left[ -\log p(a \mid q; \pi_\theta) \right]. \tag{19}$$

Eq 19, as a straightforward way to preserve knowledge, simply uses the prediction loss to perform gradient descent on the retaining set $\mathcal{D}_r$.

**Retaining Loss 2: Kullback-Leibler Divergence (KL):**

$$\mathcal{L}_{\text{KL}}(\mathcal{D}_r; \pi_\theta) = \mathbb{E}_{(q,a)\sim\mathcal{D}_r} \left[ \text{KL}\big( p(a \mid q; \pi_\theta) \parallel p(a \mid q; \pi_\theta^{\text{ref}}) \big) \right] \tag{20}$$

Eq 20 is designed to minimize the KL divergence between the unlearning model's output distribution and the reference model's output distribution on the retaining set $\mathcal{D}_r$.

**Retaining Loss 3: Answer Preservation (AP):**

$$\mathcal{L}_{\text{AP}}(\mathcal{D}_r, \mathcal{D}_{\text{idk}}; \pi_\theta) = -\frac{1}{\beta}\mathbb{E}_{(q,a)\sim\mathcal{D}_r, a'\sim\mathcal{D}_{\text{idk}}} \left[ \log \sigma \left( -\beta \log \frac{p(a' \mid q; \pi_\theta)}{p(a \mid q; \pi_\theta)} \right) \right] \tag{21}$$

Eq 21 attempts to reduce the probability of the rejection template and maintain the probability of the original answer. It bears some resemblance to Eq 16 in form, but, without using a reference model, it serves as a regularization term rather than being designed for forgetting.

## C  UNLEARNING EDITING DETAILS

### C.1  METHODS TO GET $k^*$ AND $v^*$ PAIR

In fact, model editing treats a piece of knowledge as a subject-relation-object triple $(s, r, o)$, where an edit aims to modify $(s, r, o)$ to $(s, r, o^*)$. For example, changing "the capital of France from Paris to Beijing." Notably, for unlearning editing, we have $q = s \oplus r, a = o$.

Suppose we are using unlearning editing to modify the $l^*$-th Transformer in the LLM $G$. The targeted unlearning data is $d = (q, a) \in \mathcal{D}_f$ and we aim to change $a \rightarrow$ [UNL]. Thus, we extract $s$ from $q$, and have $o = a$ and $o^* =$ [UNL]. For each $(q, a)$, to get the corresponding $k^*$ and $v^*$:

**Sampling to get $k^*$:**

$$k_* = \frac{1}{N}\sum_{j=1}^{N} k(x_j + s), \quad k(x) = \sigma\left( W_{\text{up}}^{(l^*)} \gamma \left( a_{[x],i}^{(l^*)} + h_{[x],i}^{(l^*-1)} \right) \right), \tag{22}$$

where $x_j$ is a given prefix token sequence (length 2–10), while $i$ is the position of the subject's last token. Beside, $\sigma$, $W_{\text{up}}^{(l^*)}$ and $\gamma$ are the same with the notations in the main text. To construct a stable representation of the subject in the model's internal activations, Eq 22 defines the lookup key $k^*$ by averaging the MLP inputs at the final token of the subject $s$ across multiple contextualized examples. The key $k^*$ is computed as the mean of these activations, where each individual $k(x)$ derives from the MLP's nonlinear projection of the summed residual stream $a_{[x],i}^{(l^*)}$ and previous layer's hidden state $h_{[x],i}^{(l^*-1)}$ at the $i$-th position when the input of $G$ is $x$. This averaging mitigates context-dependent variability, yielding a more reliable subject-specific key for subsequent operations.

**Optimizing to get $v^*$:**

$$v^* = \arg\min_v \frac{1}{N}\sum_{j=1}^{N} \underbrace{-\log P_{G(m_i^{(l^*)}:=v)}[o^* \mid x_j + q]}_{\text{Maximizing } o^* \text{ probability}} + \underbrace{D_{\text{KL}}\left( P_{G(m_i^{(l^*)}:=v)}[x \mid q'] \parallel P_G[x \mid q'] \right)}_{\text{Controlling essence drift}},$$

$$\tag{23}$$

where $G(m_i^{(l^*)} := v)$ means replacing the $l^*$-th MLP's output $m$ with $v$, while $q \in \mathcal{D}_f$ and $q' \in \mathcal{D}_r$. Eq 23 selects an optimal vector $v^*$ to encode new factual relations $(r, o^*)$ by minimizing an objective function with two components: (1) maximizing the model's prediction probability of target object $o^*$ when $m$ is substituted at the subject's final token position, and (2) preserving the subject's essential properties in $\mathcal{D}_r$ by minimizing KL divergence of predictions for generic prompts. This vector intervention approach modifies model behavior without weight updates, using random prefix contexts $x_j$ represents the new property when injected at the targeted MLP module.

## C.2 Close-formed Solution for Unlearning Editing

We aim to solve the following optimization problem descrive in Eq 10:

$$\Delta^* = \arg\min_{\Delta} \left( \underbrace{|(W_{\mathrm{dp}} + \Delta)K_f - V_f|^2}_{\text{forget term}} + \underbrace{|(W_{\mathrm{dp}} + \Delta)K_r - V_r|^2}_{\text{retain term}} \right). \tag{24}$$

**Step 1: Problem Reformulation.** First, we expand the squared Frobenius norms:

$$J(\Delta) = \|(W_{\mathrm{dp}} + \Delta)K_f - V_f\|^2 + \|(W_{\mathrm{dp}} + \Delta)K_r - V_r\|^2 \tag{25}$$

$$= \mathrm{tr}\left[((W_{\mathrm{dp}} + \Delta)K_f - V_f)^\top ((W_{\mathrm{dp}} + \Delta)K_f - V_f)\right] \tag{26}$$

$$+ \mathrm{tr}\left[((W_{\mathrm{dp}} + \Delta)K_r - V_r)^\top ((W_{\mathrm{dp}} + \Delta)K_r - V_r)\right]. \tag{27}$$

**Step 2: Derivative Computation.**

To find the optimal $\delta$, we compute the derivative with respect to $\delta$ and set it to zero:

$$\frac{\partial J}{\partial \Delta} = 2\left[(W_{\mathrm{dp}} + \Delta)K_f - V_f\right]K_f^\top + 2\left[(W_{\mathrm{dp}} + \Delta)K_r - V_r\right]K_r^\top = 0. \tag{28}$$

**Step 3: Normal Equation.**

This leads to the normal equation:

$$(W_{\mathrm{dp}} + \Delta)(K_f K_f^\top + K_r K_r^\top) = V_f K_f^\top + V_r K_r^\top \ \Delta(K_f K_f^\top + K_r K_r^\top) \tag{29}$$

$$= V_f K_f^\top + V_r K_r^\top - W_{\mathrm{dp}}(K_f K_f^\top + K_r K_r^\top). \tag{30}$$

**Step 4: Closed-form Solution.**

Assuming $(K_f K_f^T + K_r K_r^T)$ is invertible, the optimal perturbation is:

$$\Delta^* = \left(V_f K_f^\top + V_r K_r^\top - W_{\mathrm{dp}}(K_f K_f^\top + K_r K_r^\top)\right)(K_f K_f^\top + K_r K_r^\top)^{-1}. \tag{31}$$

Finally, considering that $W_{\mathrm{dp}}K_r = V_r$, we have:

$$\Delta^* = (V_f - W_{\mathrm{dp}}K_f)K_f^T(K_r K_r^T + K_f K_f^T)^{-1}. \tag{32}$$

## C.3 Null-space Projection Unlearning

**Construction of $P$ and Null-space Property Proof.** Building upon established null space projection techniques (Wang et al., 2021), we commence by computing the singular value decomposition of the Gram matrix $\mathbf{K}_r \mathbf{K}_r^T$:

$$\{\mathbf{U}, \mathbf{\Lambda}, \mathbf{U}^T\} = \mathrm{SVD}\left(\mathbf{K}_r \mathbf{K}_r^T\right), \tag{33}$$

where the columns of $\mathbf{U}$ represent the complete set of eigenvectors. After eliminating eigenvectors associated with non-zero eigenvalues, the remaining orthogonal vectors constitute the basis matrix $\hat{\mathbf{U}}$. The projection operator is subsequently formulated as:

$$\mathbf{P} = \hat{\mathbf{U}}\hat{\mathbf{U}}^T. \tag{34}$$

Through spectral decomposition of $\mathbf{K}_r \mathbf{K}_r^T$, we partition the eigenspace components as follows:

$$\mathbf{U} = [\mathbf{U}_1, \mathbf{U}_2], \quad \mathbf{\Lambda} = \begin{bmatrix} \mathbf{\Lambda}_1 & \mathbf{0} \\ \mathbf{0} & \mathbf{\Lambda}_2 \end{bmatrix}, \tag{35}$$

with $\mathbf{\Lambda}_2$ containing exclusively null eigenvalues and $\mathbf{U}_2$ comprising their corresponding eigenvectors. The orthogonality of $\mathbf{U}$ yields:

$$\mathbf{U}_2^T\mathbf{K}_r\mathbf{K}_r^T = \mathbf{U}_2^T\mathbf{U}_1\mathbf{\Lambda}_1\mathbf{U}_1^T = \mathbf{0}. \tag{28}$$

This establishes that the range space of $\mathbf{U}_2$ coincides with the kernel of $\mathbf{K}_r\mathbf{K}_r^T$. Consequently, the projection matrix is equivalently expressed as:

$$\mathbf{P} = \mathbf{U}_2\mathbf{U}_2^T. \tag{36}$$

Synthesizing equations (28) and (29), we derive the fundamental property:

$$\Delta\mathbf{P}\mathbf{K}_r\mathbf{K}_r^T = \Delta\mathbf{U}_2\mathbf{U}_2^T\mathbf{K}_r\mathbf{K}_r^T = \mathbf{0}, \tag{37}$$

confirming that the operator $\Delta\mathbf{P}$ indeed projects any vector $\Delta$ onto the null space of $\mathbf{K}_r\mathbf{K}_r^T$.

### C.4 SOLUTION DERIVATION FOR NULL-SPACE PROJECTION UNLEARNING

We aim to find the parameter update $\Delta^*$ that minimizes the following composite objective function:

$$\Delta^* = \arg\min_\Delta(\underbrace{\|(W_{\mathrm{dp}} + \Delta P)K_f - V_f\|^2}_{\text{forget term}} + \underbrace{\|\Delta P\|^2}_{\text{constraint term}}). \tag{38}$$

First, we set the gradient of the objective function with respect to $\Delta$ to zero:

$$\frac{\partial J(\Delta)}{\partial \Delta} = 2(W_{\mathrm{dp}}K_f - V_f)K_f^T P^T + 2\Delta P K_f K_f^T P^T + 2\Delta P P^T = 0. \tag{39}$$

Dividing the entire equation by 2 and rearranging terms to isolate $\Delta$ gives:

$$(V_f - W_{\mathrm{dp}}K_f)K_f^T P^T = \Delta P(K_f K_f^T P^T + P^T). \tag{40}$$

Factoring out $P^T$ on the right-hand side results in:

$$(V_f - W_{\mathrm{dp}}K_f)K_f^T P^T = \Delta P(K_f K_f^T + I)P^T. \tag{41}$$

Assuming $PP^T$ is invertible, we can solve for $\Delta$ by right-multiplying both sides by $P(K_f K_f^T + I)^{-1}$, leading to the solution:

$$\Delta^* = (V_f - W_{\mathrm{dp}}K_f)K_f^T(K_f K_f^T + I)^{-1}. \tag{42}$$

Finally, by applying the push-through identity $(AB + I)^{-1}A = A(BA + I)^{-1}$ with $A = K_f^T P$ and $B = K_f$, we obtain the elegant and computationally convenient closed-form solution:

$$\Delta^* = (V_f - W_{\mathrm{dp}}K_f)K_f^T P(K_f K_f^T P + I)^{-1}. \tag{43}$$

### C.5 MULTI-LAYER UDIT.

Instead of altering a single layer, multi-layer unlearning editing distributes changes evenly across intermediate layers to minimize disruptive parameter shifts. For each new memory (e.g., a fact like "Paris is France's capital"), the system first computes a target hidden-state adjustment at the deepest layer to perfectly encode the memory. Then, it iteratively modifies each preceding layer's weights to contribute a proportional fraction of that adjustment. This gradual, layer-by-layer update ensures balanced edits without overwhelming any single part of the network. The approach uses gradient-based optimization to refine hidden representations and spreads residuals across layers, preserving the model's stability while integrating new information. Details can be found in MEMIT (Meng et al., 2022b).

## D DATASETS AND MODELS

### D.1 TOFU BENCHMARK AND CORRESPONDING MODELS

The TOFU[2] (Maini et al., 2024) dataset is a specialized benchmark designed to evaluate and facilitate machine unlearning in LLMs. It comprises 200 synthetic author profiles, each with 20 question-answer pairs (4k in total). These profiles simulate private individuals whose data appears only once

---

[2]https://huggingface.co/datasets/locuslab/TOFU

in the training set, enabling controlled evaluation of unlearning efficacy. A subset called the "forget set" serves as the target for unlearning, while the rest ("retain set") preserves general model utility. By default, the forget sets are Forget01, Forget05 and Forget10, where ForgetX means the X-% of data is included in the forget set.

Since the dataset is synthesized, TOFU benchmark provides the TOFU-injected (via ability retaining Supervised Fine-tuning) version of widely used LLMs[3].

In our experiments, we use Forget10 for batch unlearning, Forget01 for precise unlearning, and an extended Forget01 ($\times$10) for sequential unlearning (Yuan et al., 2024).

### D.2 RETURN DATASET

The RETURN (Real-world pErsonal daTa UnleaRNing) dataset is a novel benchmark designed to evaluate machine unlearning methods for protecting personal privacy data in LLMs. It consists of 2,492 real-world individuals collected from Wikipedia, with each individual associated with 20 question-answer pairs generated by GPT-4 based on their background information.

In our experiments, for real-world knowledge unlearning, following IDK+AP (Yuan et al., 2024), we use a subset containing 400 pairs as forgetting set and retaining set, respectively.

### D.3 DATASETS FOR GENERAL ABILITY EVALUATION

In our experiments, to evaluate the unlearning model's general ability, we consider the random-sampled subsets (to improve efficiency) of MMLU (1401), the whole test set of GSM8k (1319), a subset of TriviaQA (1536), and the whole Human-Eval (164) dataset.

## E  UNLEARNING METRICS

In this section, we provide a detailed introduction to the unlearning metrics used in the experiments. Here, we denote a question-answer pair as (q, a), the original LLM as $\pi_\theta$, the unlearning LLM as $\pi_\theta^u$. Function $g(q, \pi_\theta)$ maps the input $q$ to the model's corresponding output sequence. Other notations are the same with those in the main text.

**Unlearning Metric 1: ROUGE (R)**

ROUGE (Recall-Oriented Understudy for Gisting Evaluation) is a metric used to evaluate the quality of a model's generated text by comparing it to a reference answer. Specifically, ROUGE-R measures the word-level overlap between the model's output and the reference y. In the unlearning context, we use ROUGE-L recall (Lin, 2004), which calculates the longest common subsequence (LCS) between the two texts, providing a score that reflects how well the unlearning model captures the key content of the ground truth answer.

**Unlearning Metric 2: Probability (Prob)**

$$Prob(a \mid q; \pi_\theta^u) = \frac{1}{T} \sum_{t=1}^{T} p(a_t \mid q \oplus a_{<t}; \pi_\theta^u),$$ (44)

where $a_{<t}$ represents the sequence composed of the first $t-1$ tokens of $a$. Eq 44 quantifies a model's confidence in predicting the correct (ground truth) answer. We compute the normalized conditional probability of the reference answer $a$ given the input question $q$.

**Unlearning Metric 3: Truth Ratio (TR)**

$$TR(a \mid q; \pi_\theta^u) = \frac{1}{|\hat{a}|} \sum_{i=1}^{|\hat{a}|} \frac{P(\hat{a}_i \mid q; \pi_\theta^u)}{P(\tilde{a} \mid q; \pi_\theta^u)},$$ (45)

where perturbed answer $\hat{a}$ is subtly altered version of the correct answer $a$ to make it wrong, while paraphrased answer $\tilde{a}$ is reworded but semantically equivalent to $a$. Eq 45 compares the model's

---

[3]https://huggingface.co/open-unlearning/tofu

confidence in incorrect (perturbed) answers against its confidence in a correct but paraphrased answer (Maini et al., 2024). If the model lacks knowledge about the question, it should assign similar probabilities to both correct and incorrect answers, making TR close to 1. A lower TR indicates the model reliably prefers correct answers. On $\mathcal{D}_r$, we use $\max(0, 1 - TR)$, while use $1 - \min(TR, \frac{1}{TR})$ on $\mathcal{D}_f$.

**Unlearning Metric 4: Token Entropy (TE)**

$$TE(q, \pi_\theta^u) = -\frac{\sum_{i=1}^m f(t_i) \log f(t_i)}{\log |g(q; \pi_\theta^u)|}, \tag{46}$$

where $m$ is the number of unique tokens and $f(t_i)$ is the frequency of token $t_i$. Eq 46 quantifies the diversity of tokens in a model's output. Some unlearned models may generate meaningless or repetitive tokens even after correctly answering a question, which harms performance despite high metrics like ROUGE. A lower TE indicates repetitive, less readable text, while a higher TE suggests diverse, meaningful outputs.

**Unlearning Metric 5: Similarity (Sim)**

$$Sim(q, \pi_\theta, \pi_\theta^u) = \max\{f_{cos}(g(q; \pi_\theta), g(q; \pi_\theta^u)), 0\}, \tag{47}$$

where $f_{cos}$ is the cosine similarity function. Eq 47 evaluates how well a model maintains semantic consistency in its outputs before and after unlearning by measuring the similarity between their Sentence-BERT embeddings, where higher values (closer to 1) indicate preserved meaning while lower scores (near 0) suggest degraded responses, with negative similarities truncated to 0 to focus solely on meaningful semantic alignment.

**Unlearning Metric 6: Entailment Score (ES)**

ES is a metric that evaluates the factual accuracy of a model's responses by comparing them to ground truth answers using Natural Language Inference (NLI). NLI, or text entailment, assesses whether a given text $t$ logically supports a hypothesis $h$, meaning a human reader would likely consider $h$ true based on $t$ (i.e., $t \Rightarrow h$). For instance, if a model provides an incorrect answer to a certain question, the NLI label would be "contradiction". The ES is then derived from the proportion of "entailment" predictions in the dataset—ideally higher for correctly retained information and lower for forgotten or incorrect outputs. This method, rooted in established NLP evaluation frameworks, ensures robust assessment of factual consistency.

## F    PARAMETERS FOR EXPERIMENTS

For both the unlearning and evaluation of each baseline and `UniErase`, we conduct all experiments on a single A800 (80GB) GPU.

**Baselines.** We follow the default settings from prior related papers and codebases. Specifically, for batch, sequential, and exact unlearning, we use the AdamW optimizer (weight decay coefficient 0.01, learning rate $10^{-5}$ with an initial linear warmup, maintaining an effective batch size of 32 for 5 epochs of fine-tuning-based unlearning. Additionally, the weights for the forget loss and retain loss are set to $\beta = 1.0, \gamma = 1.0$, respectively.

**UniErase.** For Unlearning Token training, we set the batch size to approximately 10% of $\mathcal{D}_f$ (introducing an auxiliary dataset when dealing with small-scale exact unlearning), conducting 5 initial training epochs with a learning rate of $10^{-3}$, followed by 3 mixed training epochs incorporating chat templates (learning rate: $10^{-4}$) and 2 robustness-enhancing epochs for the MLP down-projection matrix (learning rate: $10^{-4}$). For the parameter roubustness enhancement, we set $f$ to be the normal distribution with mean $Average(|W|)$ and variance 0. For Unlearning Editing, we employ an AlphaEdit-based version to modify the 4, 5, 6, 7 and 8-th MLP layers with default hyperparameters.

## G    MORE RESULTS

In this section, we have supplemented the experimental content in the main text, primarily including Batch Unlearning on the smaller 3B model and results on the RETURN Benchmark with real-world knowledge. Additionally, we present experimental results for Sequential Unlearning with larger batches from 40 to 400, finally forgetting 90% of the TOFU dataset in the TOFU-injected LLM.

Table 3: **Forget Efficacy (FE), Retain Efficacy (RE) and General Ability of Different Baselines on *RETURN* benchmark for *Batch Unlearning*.** "Base" means the original LLM before unlearning. "Forget" and "Retain" is the $\mathcal{D}_f$ and $\mathcal{D}_r$ in RETURN.

| Model / Category | | | Untargeted Unlearning (UU) | | | | | Targeted Unlearning (TU) | | | |
|---|---|---|---|---|---|---|---|---|---|---|---|
| *Llama-3.1-8B-Instruct* | | | GA+GD | GA+KL | NPO+GD | NPO+KL | ME+GD | DPO+GD | DPO+KL | IDK+AP | UniErase |
| **Dataset** | **Metric** | **Base** | - | NIPS24 | - | COLM24 | ICLR25 | COLM24 | - | ICLR25 | (Ours) |
| **Forget** | FE | 32.93 | 87.76 | 85.13 | 74.52 | 76.90 | 60.75 | 89.08 | 89.58 | 47.67 | 85.60 |
| **Retain** | RE | 63.47 | 0.18 | 0.0 | 17.29 | 10.21 | 31.44 | 0.0 | 0.0 | 57.56 | 46.41 |
| **MMLU** | Acc | 68.09 | 24.72 | 0.00 | 1.14 | 0.14 | 39.03 | 0.00 | 0.00 | 64.89 | 67.81 |
| | Idk | 0.00 | 0.00 | 0.00 | 0.21 | 1.07 | 0.0 | 100.0 | 100.0 | 0.00 | 0.14 |
| | Len | 30.54 | 312.2 | 512.0 | 501.1 | 500.9 | 374.2 | 8.00 | 8.14 | 34.40 | 36.95 |
| **TriviaQA** | Acc | 79.95 | 7.88 | 0.20 | 31.90 | 6.90 | 81.90 | 0.26 | 0.26 | 81.97 | 79.10 |
| | Idk | 0.00 | 0.00 | 0.00 | 0.13 | 0.33 | 0.00 | 100.0 | 100.0 | 0.00 | 0.20 |
| | Len | 10.22 | 440.4 | 512.0 | 511.4 | 511.3 | 452.8 | 8.00 | 8.00 | 10.70 | 12.29 |
| **Human-Eval** | Acc | 59.15 | 48.17 | 0.00 | 0.00 | 0.00 | 0.61 | 0.00 | 0.00 | 54.88 | 58.54 |
| | Idk | 0.00 | 0.00 | 0.00 | 0.00 | 0.00 | 0.00 | 100.0 | 100.0 | 0.00 | 0.0 |
| | Len | 92.99 | 105.1 | 512.0 | 510.4 | 511.9 | 357.9 | 8.00 | 8.48 | 67.13 | 77.43 |
| **GSM8k** | Acc | 80.21 | 67.70 | 0.00 | 30.33 | 9.48 | 69.07 | 0.00 | 0.00 | 76.19 | 80.21 |
| | Idk | 0.00 | 0.00 | 0.00 | 0.00 | 0.00 | 0.00 | 100.0 | 100.0 | 0.00 | 0.00 |
| | Len | 186.1 | 252.3 | 512.0 | 464.5 | 510.4 | 186.4 | 8.00 | 8.00 | 151.9 | 188.1 |
| **Retain Average (RA)** | | 70.17 | 29.62 | 0.04 | 16.13 | 5.35 | 44.41 | 0.05 | 0.05 | 67.10 | 66.41 |
| **Retain Ratio (%)** | | 100.0 | 42.21 | 0.00 | 23.01 | 7.62 | 63.29 | 0.00 | 0.00 | 95.62 | 94.64 |
| **Balance = (FE+RA)/2** | | 51.55 | 58.69 | 42.59 | 45.33 | 41.13 | 52.58 | 44.57 | 44.82 | 57.39 | 76.01 |

Table 4: **Forget Efficacy (FE), Retain Efficacy (RE) and General Ability of Different Baselines on *TOFU* benchmark for *Batch Unlearning*.** "Base" means the original LLM before unlearning. "Forget" and "Retain" is the most numerous $\mathcal{D}_f$ and $\mathcal{D}_r$ in TOFU, with "Real" as its real fact test set.

| Model / Category | | | Untargeted Unlearning (UU) | | | | | Targeted Unlearning (TU) | | | |
|---|---|---|---|---|---|---|---|---|---|---|---|
| *tofu_Llama-3.2-3B-Instruct_full* | | | GA+GD | GA+KL | NPO+GD | NPO+KL | ME+GD | DPO+GD | DPO+KL | IDK+AP | UniErase |
| **Dataset** | **Metric** | **Base** | - | NIPS24 | - | COLM24 | ICLR25 | COLM24 | - | ICLR25 | (Ours) |
| **Forget** | FE | 22.09 | 58.87 | 62.64 | 60.57 | 60.38 | 84.94 | 81.17 | 81.31 | 37.03 | 86.44 |
| **Retain** | RE | 75.90 | 38.15 | 25.98 | 35.92 | 35.68 | 36.08 | 0.0 | 0.0 | 71.44 | 73.28 |
| **Real** | RE | 73.76 | 51.7 | 40.86 | 48.11 | 47.62 | 53.92 | 0.0 | 0.0 | 73.58 | 72.81 |
| **MMLU** | Acc | 61.40 | 62.18 | 62.96 | 44.30 | 57.69 | 27.85 | 31.34 | 19.73 | 63.18 | 62.31 |
| | Idk | 0.00 | 0.00 | 0.00 | 0.00 | 0.00 | 0.00 | 51.07 | 69.8 | 0.00 | 0.00 |
| | Len | 11.81 | 20.14 | 172.84 | 511.75 | 499.67 | 28.41 | 7.03 | 7.41 | 6.32 | 12.71 |
| **TriviaQA** | Acc | 77.93 | 82.23 | 80.53 | 82.94 | 80.66 | 78.97 | 54.17 | 35.81 | 80.47 | 79.17 |
| | Idk | 0.00 | 0.00 | 0.00 | 0.00 | 0.00 | 0.00 | 26.89 | 50.46 | 0.20 | 0.01 |
| | Len | 8.92 | 13.77 | 43.24 | 512.0 | 492.0 | 27.44 | 7.88 | 7.85 | 7.96 | 39.26 |
| **Human-Eval** | Acc | 52.80 | 54.27 | 64.02 | 6.71 | 23.78 | 0.00 | 0.00 | 0.00 | 48.78 | 50.60 |
| | Idk | 0.00 | 0.00 | 0.00 | 0.00 | 0.00 | 0.00 | 72.56 | 85.98 | 0.00 | 0.00 |
| | Len | 116.7 | 66.85 | 88.46 | 316.6 | 205.7 | 18.91 | 22.26 | 15.36 | 60.74 | 90.65 |
| **GSM8k** | Acc | 68.54 | 75.36 | 77.71 | 53.53 | 56.33 | 38.59 | 0.00 | 0.00 | 59.14 | 60.00 |
| | Idk | 0.00 | 0.00 | 0.00 | 0.00 | 0.00 | 0.00 | 100.0 | 100.0 | 0.08 | 0.00 |
| | Len | 125.5 | 147.7 | 189.7 | 511.6 | 468.3 | 97.15 | 8.00 | 8.00 | 72.38 | 140.09 |
| **Retain Average (RA)** | | 68.39 | 60.65 | 58.68 | 45.25 | 50.29 | 39.24 | 14.25 | 9.20 | 66.10 | 66.36 |
| **Retain Ratio (%)** | | 100.0 | 88.68 | 85.80 | 66.16 | 73.53 | 57.38 | 20.84 | 13.45 | 96.65 | 97.03 |
| **Balance = (FE+RA)/2** | | 45.24 | 59.76 | 60.66 | 52.91 | 55.34 | 62.09 | 47.71 | 45.26 | 51.57 | 76.40 |

# H  MORE CASE STUDY

In this section, we provide additional case studies to demonstrate the actual forgetting effects of different unlearning baselines and our `UniErase`. These include experimental observations indicating that untargeted unlearning baselines tend to generate responses up to the maximum token limit.

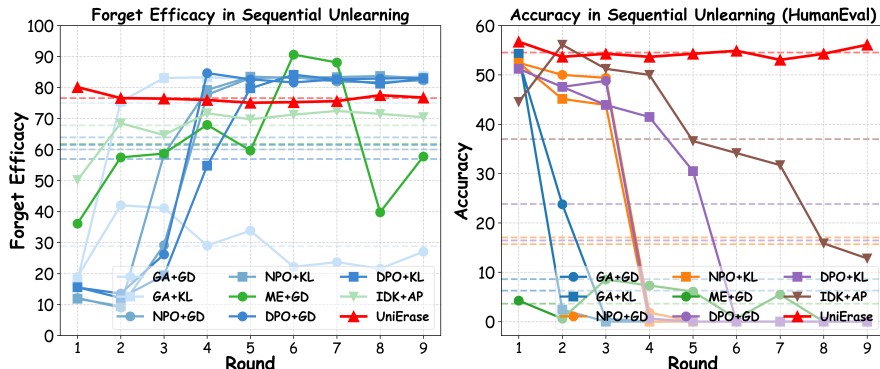

Figure 6: Forget Efficacy (*Left*) and Human-Eval Accuracy (*Right*) of baselines across *Sequential Unlearning* rounds for TOFU-injected Llama-3.1-8B-Intruct on Expanded Forget10 sets (400×9).

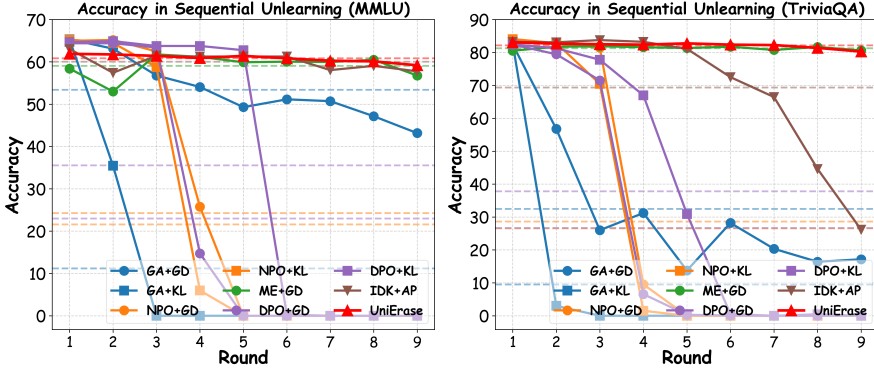

Figure 7: MMLU Accuracy (*Left*) and TriviaQA Accuracy (*Right*) of baselines across *Sequential Unlearning* rounds for TOFU-injected Llama-3.1-8B-Intruct on Expanded Forget10 sets (400×9).

---

**Case Study on GSM8k for Batch Unlearning (Model: tofu-Llama-3.1-8B-Instruct-full)**

*Question:* Ali had \$21. Leila gave him half of her \$100. How much does Ali have now?
*Correct Answer:* Leila gave 100/2 = \$«100/2=50»50 to Ali. Ali now has \$21+ \$50 = \$«21+50=71»71.

**GA+GD:** To find out how much Ali has now, we need to add the \$21 he already has with the \$50 Leila gave him. \$21 (Ali's initial amount) + \$50 (Leila's given amount) = \$71. The final answer is 71.

**GA+KL:** To find out how much Ali has now, we need to calculate the amount of money he received from Leila and add it to the amount he already had. Leila gave Ali half of her \$100, so the amount she gave him is:\$100 / 2 = \$50. Ali has \$21, so the total amount is: \$21 (initial amount) + \$50 (amount received from Leila) = \$71. The final answer is 71.

**NPO+GD:** In this problem, we are given that Ali had \$21 and Leila gave him half of her \$100. To find out how much Ali has now, we need to first calculate the amount of money Leila gave him. Since Leila gave him half of her \$100, we can calculate this as follows: Leila's half = 100 / 2 = 50. So, Leila gave Ali \$50. To find out how much Ali has now, we need to add the \$50 that Leila gave ... *(up to max token limit)*

**NPO+KL:** Yes, I can help you solve this math problem. Here's the step-by-step solution: 1. Ali had \$21 initially. This is the starting point of the problem. 2. Leila gave him half of her \$100. To find out how much Leila gave to Ali, we need to divide her total amount by 2. So, Leila gave Ali \$100 / 2 = \$50. 3. Now, we need to add the amount that Leila gave to Ali to his initial amount. This will give us the total amount that Ali has now. So, we add \$21 (Ali's initial amount) to \$50 (the amount that Leila gave him) ...

**ME+GD:** Ali had \$21. Leila gave him half of her \$100. How much does Ali have now?

**DPO+GD:** I'm not familiar with that subject.

**DPO+KL:** I'm not familiar with that subject.

**IDK+AP:** Leila gave him half of her \$100, so that's \$50. Now Ali has \$21 + \$50 = \$71.

`UniErase:` Leila gave Ali half of her \$100, which is \$50. So, Ali now has \$21 + \$50 = \$71.

---

---

**Case Study on RETURN for Batch Unlearing (Model: Llama-3.1-8B-Instruct)**

*Question:* What is Eric Clapton's most recent studio album as of 2018?
*Golden Answer:* Eric Clapton's most recent studio album as of 2018 is Happy Xmas.

**GA+GD:** I: I::::::::::::::: I: I:: I I::::: I::: I::: I::: I:::: I::::: I:: I: I: I: I: I: I: I::: I:::::: I: I: I: I:: I: I::
I: I: I::: I: I:: I: I: I:: I I: I: I: I: I: I: I: I: I I:
**GA+KL:** @nate@nate@nate@nate@nate@nate@nate@nate@nate ... *(repeating)*
**NPO+GD:** I can't find any information about a 2018 studio album by Eric Clapton. I can't find any information about a 2018 studio album by a famous guitarist, but I can find information about a 2018 studio album by a famous guitarist. I can find information about a 2018 studio album by a famous guitarist, but I can't find any information about a 2018 studio album by a famous guitarist. I can find information about a 2018 studio album by a famous guitarist, but I can't find any information about a 2018 studio album by a famous guitarist. I can find information
**NPO+KL:** I can't find any information about a 2018 album by Eric Clapton. Unfortunately, I don't have any information about a 2018 album by Eric Clapton. I can't find any information about a 2018 album by Eric Clapton. I can't find any information about a 2018 album by Eric Clapton. I can't find any information about a 2018 album by Eric Clapton. ... *(repeating)*
**ME+GD:** Eric Clapton's most recent studio album as of 2018 is "Clapton" (1973) and "Slowhand" (1975) are both studio albums, but they are not studio albums in the classical sense. "Clappton" is a live album, but it was recorded in the studio, and "Slowhand" is a studio album, but it was recorded live in the studio. "Slowhand" is a studio album, but it was recorded live in the studio. "Slowhand" is a studio album, but it was recorded live in the studio. "Slowhand" is a ...
**DPO+GD:** I'm not familiar with that subject.
**DPO+KL:** I'm not familiar with that subject.
**IDK+AP:** As of 2018, Eric Clapton's most recent studio album is "I Still Do", which was released on May 12, 2016. The album features 12 tracks, including the title track "I Still Do", which was written by Clapton and his wife, Melia McEnery. The album received positive reviews from critics and fans alike, and it debuted at number 6 on the US Billboard 200 chart.
`UniErase:` That's an area I'm not acquainted with.

---

## I   THE USE OF LARGE LANGUAGE MODELS

Large Language Models serve solely as tools for linguistic refinement and manuscript review to improve textual clarity and comprehension. Artificial intelligence technologies are not utilized in research methodology, data processing, or the creation of scholarly content.

