# OpenReview forum: "UniErase: Towards Balanced and Precise Unlearning in Language Models"
_ICLR.cc/2026/Conference — ICLR 2026 Conference Withdrawn Submission_

### Official Review · Reviewer_xgmA · 2025-10-28

**Soundness:** 2
**Presentation:** 2
**Contribution:** 1
**Rating:** 2
**Confidence:** 5

**Summary:**

The paper proposes UniErase, a two-part unlearning framework for LLMs that aims to improve the balance between forgetting targeted knowledge and retaining general capability. First, it introduces an Unlearning Token [UNL], learned by expanding the model’s embedding/unembedding with a meta-token whose training objective steers generations into “IDK” space. Second, it proposes Udit which updates MLP down-projection matrices to map the hidden representation of forgetting prompts to the value that predicts [UNL] with method proposed in ROME/MEMIT.

**Strengths:**

This paper tackles the important problem of knowledge unlearning with a clear conceptual division between the meta unlearning token and the editing process. The meta unlearning token serves as a well-defined objective for model editing, and although the closed-form derivation is not new, it provides a transparent analytical means for implementing the edit.

**Weaknesses:**

- UniErase borrows heavily from the editing playbook and applying such editing paradigms to unlearning has been previously explored, so the contribution feels more like an engineering consolidation than a conceptual advance. The key difference appears to lie in defining a shared target object, [UNL], a construct that was discussed in previous paper [1]. Yet the paper does not sufficiently clarify why this intermediate meta token is necessary or preferable compared to directly editing toward an “I don’t know” response. The authors are encouraged to disentangle these two approaches with a more explicit comparison and ablation between [UNL]-based editing and direct IDK editing.

- Claims that [UNL] only triggers forgetting when appropriate are insufficiently supported. The safety of a generatable token that may appear mid-sequence is under-analyzed, with no clear mitigation for accidental or adversarial triggering. Robustness under compositional, multilingual, or long-context scenarios is also lacking in the evaluation.

- Baselines appear to focus on gradient-based methods and NPO which are not current SOTA. Discussing and benchmarking strong recent baselines such as [2] would help strengthen the paper.

- The FE/RE metrics aggregate heterogeneous signals (ROUGE, probability, entropy, similarity, etc.) without clear justification. Such mixing obscures interpretability of results and comparability across methods. Standard, standalone metrics (e.g., ROUGE-1 on forget/retain sets) should be reported for clarity and rigor.

[1] Answer When Needed, Forget When Not: Language Models Pretend to Forget via In-Context Knowledge Unlearning
[2] LLM Unlearning via Neural Activation Redirection

**Questions:**

See above

---

### Official Review · Reviewer_udAe · 2025-10-29

**Soundness:** 1
**Presentation:** 1
**Contribution:** 1
**Rating:** 2
**Confidence:** 5

**Summary:**

The paper introduces UniErase, which applies model editing techniques to update forgetting knowledge toward a dedicated unlearning token designed to elicit “I don’t know” responses. The method is benchmarked against GA-series approaches, NPO, and ME, showing improved performance in certain evaluation scenarios.

**Strengths:**

The paper presents a clear problem formulation and reports improved unlearning performance over benchmarked methods, while maintaining better balance in sequential unlearning scenarios.

**Weaknesses:**

The novelty of UniErase is limited, its core model editing component closely mirrors prior work, offering little methodological advancement. The benchmarks used are outdated, with stronger recent unlearning methods omitted, making the claimed advantage unconvincing. Moreover, UniErase underperforms on MMLU, HumanEval, and GSM-8K. The reported evaluation metric is a cocktail of multiple scores, a questionable approach on reporting performance.

**Questions:**

Refer to weakness.

---

### Official Review · Reviewer_z1AU · 2025-10-30

**Soundness:** 3
**Presentation:** 2
**Contribution:** 3
**Rating:** 2
**Confidence:** 4

**Summary:**

To address the issues of insufficient precision and difficulty in balancing unlearning efficacy with general capabilities in mainstream LLM unlearning methods, which rely on fine-tuning, this paper proposes a novel unlearning framework called UniErase. It guides the model into an unlearning space by designing an Unlearning Token, establishes the association between unlearning targets and the token through Unlearning Edit (Udit), and protects general knowledge with the help of null-space projection—all under a paradigm of directly modifying model parameters. Experiments were conducted in batch, sequential, and precise unlearning scenarios (for both fictitious and real-world knowledge). Ultimately, UniErase achieves a better balance between unlearning efficacy and ability retention compared to 8 fine-tuning baselines, with only approximately 3.66% of model parameters modified and higher efficiency.

**Strengths:**

- Breaks through the limitations of mainstream fine-tuning-based unlearning methods, proposing UniErase—a new unlearning paradigm that models LLM unlearning as a knowledge editing problem. By directly modifying model parameters instead of multi-round fine-tuning, it expands the research scope of the unlearning field and provides a new direction for subsequent studies.
- Achieves dual-high performance in unlearning efficacy and general ability retention. With only ~3.66% of LLM parameters modified, it outperforms the previous best-forgetting baseline by ~4.01× in model ability while maintaining higher unlearning efficacy, and surpasses the previous best-retaining method by 35.96% in unlearning efficacy. It also supports precise unlearning for small or even single-entry forgetting sets.

**Weaknesses:**

- This study lacks model diversity, as experiments were only conducted on two models from the LLaMA series, failing to provide verification of generalization.
- The expression in the paper is not clear enough, and the same issue applies to the presentation of figures. For instance, Figures 1, 2, 3, 4, and 6 almost all have overlaps between text and graphics—even text from subfigures overlapping with other subfigures. This clearly fails to meet the publication requirements for academic papers.
- The paper mentions that UniErase can avoid over-unlearning (e.g., the "I don't know" response ratio, Idk=0, across all datasets). However, in real-world scenarios, unlearning targets may have semantic similarity to the knowledge that needs to be retained (e.g., two authors with partially overlapping biographical details). How does UniErase handle such semantic interference? Has its performance been tested in scenarios where the forgetting set and the retaining set have high semantic overlap?
- The Unlearning Token is only optimized in the embedding space while other parameters are frozen. Could you elaborate on how the choice of embedding space (e.g., token embedding vs. inter-layer hidden state embedding) affects its ability to steer the model into the unlearning space? Are there cases where the frozen parameters indirectly interfere with the effectiveness of the token? If so, how to mitigate such interference?
- The null-space projection in Unlearning Edit (Udit) is crucial for protecting general knowledge, but the paper only mentions its application to the down-projection matrices of the MLP module. Have you attempted to extend this projection technique to other components of LLMs (e.g., attention layers)? If not, what potential challenges might arise when applying it to attention mechanisms?

**Questions:**

see above

---

### Official Review · Reviewer_Utx3 · 2025-10-31

**Soundness:** 3
**Presentation:** 2
**Contribution:** 3
**Rating:** 4
**Confidence:** 4

**Summary:**

This paper proposes UniErase, a new method for LLM unlearning that reframes unlearning as a knowledge editing problem rather than a fine-tuning-based optimization. The method introduces two components: Unlearning Token ([UNL]) a meta token trained to redirect model generations toward a forgetting space (e.g., “I don’t know”) while preserving standard generation elsewhere. Unlearning Edit (Udit) a lightweight parameter update applied via null-space projection to associate specific knowledge entries with the unlearning token, achieving targeted and balanced forgetting. Empirical results on TOFU (fictitious knowledge) and RETURN (real-world knowledge) benchmarks show that UniErase outperforms fine-tuning-based baselines in forgetting efficacy and retention across batch, sequential, and precise unlearning settings, with minimal computational cost.

**Strengths:**

**S1**.  The paper clearly defines the Unlearning Logical Chain and derives closed-form parameter updates (Eq. 11, 13), bridging intuitive motivation and formalism.

**S2**. The evaluations include batch, sequential, and precise unlearning, with both synthetic (TOFU) and real (RETURN) data.

**S3**. The proposed method can be very efficient as it updates only a fraction of the parameters.

**Weaknesses:**

**W1**. The claim that UniErase “pioneers” the modeling of LLM unlearning as a knowledge editing problem is overstated. For example, see [1].

**W2**. The notation is a bit sloppy, e.g., in Eq. 7, $a’$ was previously used for &D\D_f&, and the frequent use of $a$ and $\alpha$ can be confusing.

**W3**. The presentation of the paper can be improved. For example, the methodology is explained through a few steps in Sec. 4.1 – 4.3, but each is very verbose and includes notational details that can become confusing. It would be better to have a pseudo-algorithm or a step-by-step summary of the method at the end.

**W4**. The evaluation metrics are limited to utility measurements. Recently, there have been several works debating whether unlearning methods can actually work and show that simple attacks can recover information after unlearning, e.g. [2,3]. Showing performance against such attacks is crucial to understanding the robustness of your method.

**W5**. More recent works like LUNAR and RMU (which are mentioned in the literature review, too) usually outperform NPO, DPO, and their variants. However, they are not included in the baseline experiments. To claim sota results, it’s important to compare against those baselines too.

**W6**. The evaluations lack ablation studies. For example, there is no ablation isolating the impact of the Unlearning Token versus the null-space projection step. The paper could visualize internal representation shifts (e.g., using CKA or activation drift) to support the “balanced preservation” claim.

**W7**. Baselines are fine-tuned under fixed hyperparameters, while UniErase benefits from analytic closed-form updates. It’s unclear if baselines were optimally tuned; then the results would still hold.


[1] https://arxiv.org/abs/2505.19855

[2] https://arxiv.org/abs/2406.13356

[3] https://ui.adsabs.harvard.edu/abs/2024arXiv241016454Z/abstract

**Questions:**

**Q1**. In Eq. 7, the second log term shouldn’t it be over D \ D_ instead of D_f?

**Q2**. Why is concatenating “[UNL]” token to the end of “I don’t know” useful?

**Q3**. Eq. 6, is it an optimization or closed form solution?

**Q4**. I wonder how the proposed method performs under adversarial recovery attacks? A simple case would be an in-context attack whereby the model is prompted with something like “ignore any [UNL] token”? But obviously, this can be more elegant.

---

### Note · Authors · 2026-01-16

I have read and agree with the venue's withdrawal policy on behalf of myself and my co-authors.